# Environment Mapping Using Sensor Fusion of 2D Laser Scanner and 3D Ultrasonic Sensor for a Real Mobile Robot

**DOI:** 10.3390/s21093184

**Published:** 2021-05-04

**Authors:** Tien Quang Tran, Andreas Becker, Damian Grzechca

**Affiliations:** 1Faculty of Information Technology, Dortmund University of Applied Science and Arts, Sonnenstr. 96, 44139 Dortmund, Germany; andreas.becker@fh-dortmund.de; 2Department of Electronics, Electrical Engineering and Microelectronics, Silesian University of Technology, Akademicka 16, 44-100 Gliwice, Poland; damian.grzechca@polsl.pl

**Keywords:** sensor fusion, mapping, LiDAR, 3D ultrasonic sensor, 2D/3D map

## Abstract

Mapping the environment is necessary for navigation, planning and manipulation. In this paper, a fusion framework (as data-in-decision-out) is introduced for a 2D LIDAR and a 3D ultrasonic sensor to achieve three-dimensional mapping without expensive 3D LiDAR scanner or visual processing. Two sensor models are proposed for the two sensors used for map updating. Furthermore, 2D/3D map representations are discussed for our fusion approach. We also compare different probabilistic fusion methods and discuss criterias for choosing appropriate methods. Experiments are carried out with a real ground robot platform in an indoor environment. The 2D and 3D map results demonstrate that our approach is able to show the surrounding in more details. Sensor fusion provides a better estimation of the environment and the ego-pose whilst lowering the necessary resources. This gives the robot’s perception of the environment more information by using only one additional low-cost 3D ultrasonic sensor. This is especially important for robust and light-weight robots with limited resources.

## 1. Introduction

Solutions for mapping problems for indoor and outdoor robots were first developed in 1989 [1]. Since then, they have played a big role in, e.g., the automotive industry [2]. Maps are important for navigation, manipulation, obstacle avoidance or planning tasks. Making a map usually requires solving a simultaneous localization and mapping (SLAM) problem, where mapping and localization run in parallel and are dependent on each other. Previous research has studied broadly the use of 2D LiDAR for mapping. One of the challenges when using 2D sensors is navigating around drop-offs or cliffs, also known as negative obstacles. Another challenge is recognizing obstacles that are out of view of 2D sensors. With only a 2D laser scanner, it makes those challenges much more difficult to be addressed. A bulky and expensive 3D laser scanner is often not a feasible solution. The key question is how to observe the whole environment without the need of an expensive 3D range scanner or image processing. The major problem with this kind of approach is that visual information may not work in some specific cases. In this paper, a fusion framework is proposed to create a map of the environment using a conventional 2D laser range sensor (SICK LMS100) and a low cost 3D ultrasonic sensor (Toposens TSAlpha). Each sensor framework has different advantages and disadvantages that will be discussed in Section 2. This study systematically reviews the techniques for sensor fusion, aiming to provide a method for improving 2D and 3D map representations. The methodological approach taken in this study is a mixed methodology based on studying and comparing different data fusing techniques, map representations and sensor models.

### 1.1. Outline

This paper is divided into four sections. Section 1 introduces the objective of this study and its related literature. Section 2 begins by laying out the theoretical foundations of the research, and looks at how to make a combined map. The sensors characteristics are first examined. After that, many data fusion methods are studied. Then the most suitable fusion method is chose based on those results. Finally, a mapping framework is introduced to fuse both sensors by making use of two inverse sensor models. Additionally, different map representations are introduced. The remaining part of the paper proceeds as follows: Section 3 introduces the solution architecture. Section 4 presents the results of mapping in 2D/3D fusing the two sensors. Last but not least, Section 5 and Section 6 discusses the result in detail and possible future developments.

### 1.2. Related Work

Different approaches exist in the literature regarding fusion of sensors with different characteristics such as camera and LiDAR. Miura et al. in [3] fused a stereo camera and a LiDAR for making a combined map by using an integration rule on every obstacle. Danfei et al. in [4] used deep neural networks to fuse 3D point clouds and image data for 3D object detection. Kang et al. [5] proposed a solution for fusing a 2D laser and camera color information for making a 2D map for autonomous vehicle. The authors of [6] fused LiDAR and radar data for object detection applications. Such approaches is limited when visual information is not accessible.

Galvez et al. [7] implemented a grid fusion with multiple 2D laser range sensors for mapping and obstacle detection. Authors of [8] used two 2D ultrasonic sensors with different spatial setup for 3D mapping. However this approach requires a 3DOF robot that can control its height such as drone. In [9], the authors proposed a 3D mapping method using a 2D camera and a 3D Photonic Mixer Devices camera. The authors in [10] introduced a fusion method using Superbayesian Independent Opinion Pool for a forward range sensor model for updating the occupancy grid map instead of a traditional inverse sensor model. They also provide an example of making an occupancy grid using two different 3D range sensors.

3D ultrasonic sensor platforms are relatively new to the market. Therefore, there are few scientific works to express its usability. To the best of our knowledge, there is no other literature or open source implementation that uses a fusing technique between two 2D/3D range sensors for mapping purposes, especially by using a robust low-cost 3D sensor to elaborate the understanding of the environment.

## 2. Materials and Methods

In this section, the material is addressed by discussing the platforms we are using. Then a methodology overview will also be conducted.

### 2.1. Platform with Sensors Applied

A Heros 444 robot platform from German Company Innok Robotics [11] is used. The robot has a skid steer drive and can be modulary configured. Our Innok Heros robot platform is equipped with a SICK LMS100 laser range sensor which returns the measurements in 2D plane (z=0) with good resolution, precision and far range. This robot is also equipped with a light-weight 3D ultrasonic sensor TS Alpha from Toposens [12] which returns 3D information with comparable high uncertainty, low resolution and a relatively short range. This raises up the question how to find the best way of a good integration of both sensors into a single map representation of the surrounding. In the following, the advantages and disadvantages of both sensors are analyzed.

#### 2.1.1. LiDAR

A 2D SICK LMS100 laser scanner [13] returns a good range measurement and resolution. This results in detailed information about the 2D plane in which the sensor is mounted. The principle of this laser sensor is that it uses a spinning mechanism to direct the laser beam. The sensor measures the time-of-flight (TOF) for range calculation.

LiDAR is often used in robotics and the automotive industry, especially for applications like mapping and navigation due to its fast update rate, precise measurements and comparable simple interpretation of data. For mapping, LiDAR is often used as reference sensor.

#### 2.1.2. 3D Ultrasonic Sensor

The TS Alpha ultrasonic sensor (US) from Toposens [12] is used (Figure 1). This 3D US uses an array of microphones as receiver [14]. This allows the sensor to compute the distance and angle of arrival of targets causing echos, hence determine the 3-dimensional coordinate of each echo object. The sound wave is created as a cone-shape by the transducer in a defined frequency. The 3D US, on the contrary to LiDAR, returns a uncertain and sparse information about the 3D surrounding. Due to physical characteristics of sound, the resulting data is effected strongly by surrounding noise or the material of the targets. Another disadvantage of the sensor is that it can not detect walls as lines in space due to the dominant specular reflection [15]. For that reason, a laser scanner for map making purposes is still used. Laser also significantly improves the position estimate if used in SLAM [16]. However, the TS alpha is a light-weight, low cost sensor. The sensor consumes very low power. Therefore, it has a great benefits in many applications. This results in the framework proposed in Section 3 that uses a 2D-SLAM for ego-pose estimation and the uses the resulting position estimate for creating a 3D-map.

#### 2.1.3. Comparison

Figure 2 shows an exemplary measurement of both sensors. As in the Figure, US measurements (green boxes) can report height information. However, comparing to the LiDAR sensor, which can only measure the 2D space (white boxes), the US returns much less measurements.

The differences between LiDAR and 3D Ultrasound are highlighted in Table 1:

### 2.2. Sensor Fusion

We are interested in how the data should be combined on different levels as mentioned in [17,18,19,20]. Three main types of fusion methods have been analyzed:Data level fusion: At this fusion level, the data should be homogeneous. The sensor measurements are taken directly and processed by a signal processing unit. It is typically solved by classical detection and estimation. In our application, even though two sensors report range measurements, however, the measurement spaces are different (2D/3D). Therefore, only considering data level fusion will most likely result in an ill-defined algorithm.Feature level fusion: If the data from different sensors are heterogeneous, it should be fused at this level. Before applying fusing approaches, representative features have to be extracted. The features are combined in either a quantitative or qualitative manner. This involves clustering, neural networks, segmentation or basic classification methods [21,22,23]. For good results, sensors should contain rich feature information like cameras or high resolution like most LiDAR sensors. Approaches like in [24,25,26] defined a solution for extracting features using point clouds as input. However, due to sparse US measurements, and 2D measurements of the LiDAR, feature detection in our case is most likely not feasible.Decision level fusion: After the sensors have made the decision (e.g., position, objects’ attributes, identity, maps or localization) independently, the fusion algorithm combines the decisions of each individual sensor to have a final decision. Example of decision method can be pooling techniques [27], Bayesian inference, and Dempster-Shafer’s method [28].

One can choose different fusion methods for different applications, based on different mentioned challenges in [29] and low to high level sensor fusion in [17]. In addition, Varshney [30] mentioned six possible combinations from three-level hierarchical of data fusion method. Table 2 represents I/O modes using input and output as fusion methods.

### 2.3. Opinion Pooling

Having studied what is meant by different fusion levels, we will now move on to discuss the methods. We decided to fuse the sensors’ decisions, which is the highest level of fusion. Each range measurement is modeled as an occupancy probability of a grid which will be described later in Section 2.6 and Section 2.7. Firstly, the occupancy map probability with multiple sensor data that is defined as follows:(1)$$p\left(M\right)=p\left(M|z_{1:t}^{1}...z_{1:t}^{i},x_{1:t}\right),$$
where:
*i*= sensor index,*M*= map of the environment,$z_{1:t}^{i}$= measurements from ith sensors,$x_{1:t}$= estimated robot’s positions.

Assuming a given position, the relation of map posterior and the probability of one single grid $m_{c}$ to be occupied at time step *t* is defined as follows:(2)$$p\left(M\right)=\prod_{c}p\left(m_{c}|z^{1:i}\right).$$
This posterior shows the likelihood of the grid being occupied, assuming that the grid is reachable from every sensor. For each sensor a probabilistic model is used (see Figure 3). This probabilistic model is later called inverse sensor model (ISM) in Section 2.6. According to the methods summarized by Dietrich et al. in [27] and Thrun in [34], there are multiple approaches for calculating this posterior:

Linear Opinion Pool (LOP) [27]: A popular combination rule is LOP, which is used by Adarve et al. in [35] for fusing multi-layer LiDAR and stereo vision system. This applies weighted average between multiple observation sources. Where the $0\leq w_{i}\leq 1$, to underline the contribution of each sensor. The linear average can be generalized to take the weighted form as:
(3)$$w_{1}p_{1}+w_{2}p_{2}+...w_{i}p_{i},$$
so the grid posterior with α as a normalizing factor becomes:
(4)$$p\left(m\right)=\alpha \sum_{i}w_{i}p_{i}\left(m\right),\;\alpha ={\left[\sum_{i}w_{i}\right]}^{-1}.$$Logarithmic Independent Opinion Pool (LIOP) or Geometric pooling [27]: The weights are summed up to unity $\sum_{i}w_{i}=1$ and $0\leq w_{i}\leq 1$. However, different from linear averaging, geometric pooling takes weighted form as:
(5)$$p_{1}^{w_{1}}p_{2}^{w_{2}}p_{3}^{w_{3}}...p_{i}^{w_{i}},$$
Firstly, it takes the geometric average of the individuals’ map posterior. Secondly, it re-normalizes these collective probabilities to become unity. The posterior with β as a normalizing factor becomes:
(6)$$p\left(m\right)=\beta \prod_{i}p_{i}{\left(m\right)}^{w_{i}},\;\beta ={\left[\prod_{i}p_{k}{\left(m\right)}^{w_{i}}+\prod_{i}{\left(1-p_{k}\left(m\right)\right)}^{w_{i}}\right]}^{-1}.$$Independent Opinion Pool (IOP) or Multiplicative pooling [27]: This method also determines the posterior in two steps. The first step is to multiply all the individual probabilities of that sensor readings together to form a collective probability rather than geometric average. In the second step, the function re-normalizes the resulting probability so that the total sum becomes unity. The final collective occupancy grid likelihood is:
(7)$$p\left(m\right)=\beta \prod_{i}w_{i}p_{i}\left(m\right)\;\beta ={\left[\prod_{i}p_{i}\left(m\right)+\prod_{i}\left(1-p_{i}\left(m\right)\right)\right]}^{-1}.$$Bayes Filter uses Bayes theorem to estimate the posterior based on a recursive manner. Equation (Equation 2) after applying Bayes theorem can be written as follows:
(8)$$p\left(m|z^{1:i}\right)=\frac{p\left(z^{1:i}|m\right)p\left(m\right)}{p(z^{1:i})}.$$
Assuming that the sensor measurements $z^{i}$ are independent:
(9)$$p\left(z^{1:i}|m\right)=\prod_{i}p\left(z^{i}|m\right).$$
Then Equation (Equation 8) becomes:
(10)$$p\left(m|z^{1:i}\right)=\frac{p(m)}{p(z^{1:i})}\prod_{i}p\left(z^{i}|m\right),$$
Applying Bayes theorem again in Equation (Equation 10):
(11)$$p\left(m|z^{1:i}\right)=\frac{p(m)}{p(z^{1:i})}\prod_{i}\frac{p\left(m|z^{i}\right)p\left(z^{i}\right)}{p(m)},$$
Equation (Equation 11) expresses occupied cell. By analogy, free cell posterior is stated as follows:
(12)$$p\left(\bar{m}|z^{1:i}\right)=\frac{p(\bar{m})}{p(z^{1:i})}\prod_{i}\frac{p\left(\bar{m}|z^{i}\right)p\left(z^{i}\right)}{p(\bar{m})},$$
The odds are calculated by taking Equation (Equation 11) divided by Equation (Equation 12):
(13)$$\frac{p(m|z^{1:i})}{p(\bar{m}|z^{1:i})}=\frac{p(m)}{p(\bar{m})}\prod_{i}\frac{p\left(m|z^{i}\right)p\left(\bar{m}\right)}{p\left(\bar{m}|z^{i}\right)p\left(m\right)},$$
Using log-odd notation to simplify the calculation for Equation (Equation 13):
(14)$$l\left(m\right)=\log\frac{p(m|z^{1:i})}{p(\bar{m}|z^{1:i})}=l\left(m\right)-Nl\left(\bar{m}\right)+\sum_{i}l\left(m|z^{i}\right),$$
where: *N* = number of sensors.If the prior knowledge of the grid is unknown, then p=0.5, in other words l(m)=0 and $l(\bar{m})=0$. The fused probability of an occupied cell given *i* sensors is then:
(15)$$l\left(m\right)=\sum_{i}l_{i}\left(m|z^{i}\right).$$There are other approaches like supra-Bayesian Independent Opinion Pool, aggregation of imprecise or qualitative probabilities [19,27]. Another method called Dempster-Shafer theory of evidence is used by Rombat et al. [28]. Maximum probability of occupancy method for every grid cell is also mentioned in [34].

The methods discussed above imply how one grid occupancy possibility will be updated within a multi-sensor fusion scenario. In the next sub-section, we will discuss the fusion methods using two specific use cases.

### 2.4. Methods Comparison

We believe there is no ideal method to rate fusion techniques. Therefore, in order to choose the right method for our application, the approaches is compared based on three performance criteria:Discarding non-informative information: The map’s belief should only follow the sensor that gives measurements. I.e., in case of single sensor contribution (i.e., due to field of view limitation of the other sensor) this sensor shall dominate the map.Supporting measurements: The belief should increase when two sensors report a same measurement.Conflicting measurements: The belief should converge to unknown state when sensors report conflicting information.
Based on those criteria, different approaches are analyzed how it affects different scenarios. Two common use cases are accessed:Single-sensor contribution: Only one sensor detects the occurrence of that cell.Multi-sensor contribution: The cell can be reached by multiple sensors. All sensor’s measurements contribute to the decision output.

At the case where cells can only be detected by one sensor. This would demonstrate the first criteria, how the fusion system deals with non-informative information. Our project uses two sensors with different spatial characteristic, e.g., one can measure height information and one cannot. Therefore this criteria is also the most important. Assuming a test scenario where all sensors return $p_{i}=0.5$, only sensor number one returns the measurement at that cell $p_{1}\neq 0.5$. We evaluate p(m) in case of equal weight ($\alpha =\frac{1}{N}$, N is number of sensors) LOP, LIOP, IOP and Bayes filter respectively:(16)$$\mathrm{LOP}:p\left(m\right)=\alpha \sum_{i}w_{i}p_{i}\left(m\right)=\frac{p_{1}\left(m\right)}{N}+\frac{(N-1)\cdot 0.5}{N}.$$
(17)$$\mathrm{LIOP}:p\left(m\right)=\frac{p_{1}{\left(m\right)}^{\frac{1}{N}}}{p_{1}{\left(m\right)}^{\frac{1}{N}}+{\left(1-p_{1}\left(m\right)\right)}^{\frac{1}{N}}}.$$
(18)$$\mathrm{IOP}:p\left(m\right)=\frac{p_{1}\left(m\right)\cdot 0.5^{N-1}}{p_{1}\left(m\right)\cdot 0.5^{N-1}+\left(1-p_{1}\left(m\right)\right)\cdot {\left(1-0.5\right)}^{N-1}}=p_{1}\left(m\right).$$
(19)$$\mathrm{Bayes}\;\mathrm{filter}:l\left(m\right)=l_{1}\left(m\right)=\frac{p_{1}\left(m\right)}{1-p_{1}\left(m\right)}.$$

As in Equations (Equation 16) and (Equation 17), the LOP and LIOP map possibility is changed by the number of sensors (dependent on N). IOP and Bayes filter on the other hand are not affected by the other sensor (independent of N) rather than the informative sensor. In conclusion, only IOP and Bayes filter are fulfilling the first criteria, where the fusion system must only be influenced by the sensor that returns information.

Secondly, the later use case is tested when two sensors address a distinct cell probability. A brute-force check is applied to analyze the influence of each sensor according to different pooling algorithm. We made two assumptions for the test case. The first assumption is that we only have two sensors: (1) representing the US and (2) representing the laser sensor. The second assumption is the first sensor (ultrasonic sensor) will have less of the contribution compared to the second sensor (laser), therefore less weight. This underlines the fact that the ultrasonic sensor data is noisier and sparser. We plot the output of final map probability p(m) which is the combination of $p_{1}\left(m\right)$ and $p_{2}\left(m\right)$ (Figure 4). This illustrates the joint probability using both sensor readings given in a contour map. Both IOP and Bayesian methods yield the same output. The differences are with Bayesian approach is that log odds notation is used and we can take into account the prior knowledge.

Along the z-axis, only the ultrasonic sensor reports information. Therefore, the fusion algorithm must ignore all the non-informative data from the LiDAR sensor. According to the first use case, where single-sensor contribution is accessed. Bayes filter and IOP can discard non-informative data and result in a map depending only one sensor (Equations (Equation 18) and (Equation 19)). On the x-y-plane, both sensors return measurements. Due to the fact that the 3D ultrasonic sensor returns several ghost and false measurements, our fusion algorithm must be able to deal with supporting information and conflicting information. All four mentioned methods can deal good with supporting information from both sensors. However, LIOP and LOP result in an influenced decision based on weighting factors. As in Figure 4, the trust area of $p_{2}\left(m\right)$ is much larger when the sensor report a high probability measurement in LIOP case. The joint likelihood will dramatically drop when $p_{2}\left(m\right)$ drops in LIOP case, too. This weighting factor is favoured when we have different preciseness in our sensor system. However, LIOP and LOP do not work properly for the first use case which is non-informative information. Therefore, those opinion pooling methods are not suitable in our application. Additionally, Bayes filter requires less computational resources compared to IOP case due to its log-odd representation. From the previous discussion, it can be seen that choosing Bayes filter approach complies with all mentioned criteria.

### 2.5. Ultrasonic Message Filtering

Sensor readings from TS Alpha are pretty sparse and noisy (see Figure 5a). Toposens provides the users the following settings for tuning their signal processing:Number of Waves: Setting the duration of the ultrasound wave sent out by TS Alpha. Shorter waves allow to better separate objects, while longer waves allow longer measurement range.Thresholding: The threshold determines how strong the reflection needs to be to be considered as an object. Firstly, the sensor is calibrated in the beginning of any frame by taking the mean ambient noise. This will take the current surrounding noise as the base line for measurements. The threshold level is the difference in reflected signal amplitude from the ambient noise in each frame.Filtering: This setting impacts target separation in case of objects with small and large echo being close to each other. A small window allows to detect small objects close to large objects whereas a large window reports both as one object. The latter reduces the number of ghost objects.Clustering function: Clustering multiple measurements to create one single object. This object is visualized as markers or point cloud data.

Figure 5a shows the points return from raw US measurements and Figure 5b shows the points after being processed. As can be seen in Figure 5a, the raw measurements from the ultrasound contain different clusters and ambiguous measurements. This will result in a map with false positive and false negative occupancy. A heuristic tuning procedure is performed to reduce as much noise as possible while remaining true positive sensor measurements. The parameters provide a trade off between range and resolution. To put it another way, through heuristic tuning, we attempt to balance between the data rich/information poor and information rich/data poor dilemma. After applying a threshold to the returning signal, a filtering step is applied to determine if it is a reflection object. In the last step, the driver allows us to cluster the points into one distant object. In the end, a set of parameters is applied which is most likely reasonable for mapping. Despite of less measurements in Figure 5b, the sensor is less prone to measurement noises and ambiguous points.

### 2.6. Map Representation

Before employing these theories to examine sensor fusion, it is necessary to discuss several map representations and the map update techniques in more detail.

#### 2.6.1. 2D Map

Occupancy grid map [1] works with noisy and uncertain sensor readings. It divides the 2D continuous space into grids. Grid presentation is widely used in robotics and self driving car technology because it uses statistics to represent the environment. Occupancy of a cell is initialized as $p_{unknown}=0.5$ (grey). Then it is represented as a binary random variable with $p_{free}=0$ (light grey), $p_{occupied}=1$ (black) and cells are independent from each other (Figure 6). A 2D map can be made by any 2D range sensor. Depending on the sensor model, the resulting map quality will vary.

In robotics, SLAM problems often use such a map representation. Approaches like gmapping [16], hector mapping [27], or google catographer [36] provide different solutions for localization and map update, however, the resulting map always follows occupancy grid representation.

#### 2.6.2. 3D Map

3D map representations are evaluated through three main categories:Probabilistic representation: Cover uncertainties and error in range sensor.Modelling of unmapped areas: For navigation and planning behaviour to explore unmapped areasEfficiency: Memory consumption, accessibility.

Making the map using normal grid has the major drawback that this representation requires large amounts of memory and thus they are not efficient to represent 3D environments. Another drawbacks is that occupancy grids must have an initial grid size. This is challenging for 3D grids due to the scope of 3D environment. Furthermore, the higher the resolution that shall be achieved is, the more memory is required. Another approach is storing the point clouds from the sensor directly and use it in SLAM systems like in [37]. But this requires a very high precision 3D sensor and cannot deal with noises and dynamic objects in the environment.

Oc-tree data (Figure 7) structure is one of the baselines for storing such a 3D map. Oc-tree data structure divides the space into voxels in a recursive manner, children voxels will have a parents voxel. One voxel is divided into eight sub voxels, therefore the tree can be expressed using parents to represent different local map resolutions. This creates a hierarchy of octants that can be easily stored and accessed. First 3D map that use Oc-tree data structure is presented by Meagher in 1982 [38]. Similar approaches can be found in [10,39]. However, these approaches have not addressed map efficiency compression method.

Octomap [40] uses oc-tree as storing method that fulfills the three mentioned categories. The authors provide a 3D map making framework using oc-tree representation using clamping update and pruning technique for memory efficiency.

The occupied voxels are updated using:(20)$$P\left(n|z_{1:t}\right)=\left[1+\frac{1-P(n|z_{t})}{P(n|z_{t})}\frac{1-P(n|z_{1:t-1})}{P(n|z_{1:t-1})}\frac{1-P(n)}{P(n)}\right].$$

P(n|z) is the inverse sensor model. Using log odds notation, the equation above becomes:(21)$$L\left(n|z_{1:t}\right)=L\left(n|z_{1:t-1}\right)+L\left(n|z_{t}\right).$$

However, when applying Equation (Equation 21) for updating the voxel state, the upper bound and lower bound is not defined. Therefore, when a voxel is observed free for n− times, that specific voxel must be hit by the sensor reading n−times to recall the same likelihood. Yguel et al. in [41] proposed an update policy that have upper ($l_{max}$) and lower bound ($l_{min}$) for Equation (Equation 21):(22)$$L\left(n|z_{1:t}\right)=max\left(min\left(L\left(n|z_{1:t-1}\right)+L\left(n|z_{t}\right),l_{max}\right),l_{min}\right).$$

This results in a faster way to update the space. This also allows the voxels to reach a stable state when the probability reaches $l_{max}$ or $l_{min}$. With this stable state, the voxel is listed as free or occupied. When all the eight children of a voxel are in a stable state, the children voxels are pruned, which leads to a compressed way to store a map.

### 2.7. Sensor Models

The grid map makes two assumptions of the world namely that the world is static and that individual grid cells are independent. The grid map is updated using an inverse sensor model (ISM) denoted as $p(m|z_{t})$.

The laser inverse sensor model is used in many occupancy mapping problems. Figure 8 shows a typical laser ISM:

This laser ISM is divided into three parts, the model for each part is denoted as follows:Before hitting obstacles, there is a low probability that there is any obstacle ($p_{free}=0.4$ is chose) (high true negative rate).Right at the measurement, the probability that there is an obstacle is very high (large true positive rate). The high probability maintains shortly depending on the grid cell size (r) ($p_{occupied}=0.7$ is chose).Beyond the measurement the probability returns to initial value ($p_{unknown}=0.5$ is chose).

The forward sensor model of the 3D ultrasonic sensor can be represented as two components that have Gaussian noise in x-y-plane and z-direction:(23)p(z(x−y)|m)=1σ(x−y)2πe−x−μ(x−y)2−x−μ(x−y)22σ(x−y)2σ(x−y)2.
(24)p(z(z)|m)=1σ(z)2πe−x−μ(z)2−x−μ(z)22σ(z)2σ(z)2.

Based on the characteristics of a forward model, we get the 3D ultrasonic ISM for updating the cell in 3D. For simplicity, we only consider along the measurement beam. A typical 3D ultrasonic ISM updating the cell *m* can be modeled as shown in Figure 9. For more details see [6].

To update the three dimension grid cells, we found through heuristic tuning, similar parameters as 2D laser should be applied to achieve a fair result:Cells are updated along the line between measurement and sensor as free space ($p_{free}=0.4$ is chose)Right at the measurement: The cell is modeled as occupied ($p_{occupied}=0.7$ is chose)Beyond the measurement: The ISM returns the prior, which is unknown grid cell ($p_{unknown}=0.5$ is chose).

In order to build the map, the ISM of each sensor is taken. After that, the occupancy is calculated by using those ISMs. Finally, since the cells are considered to be independent, fusing of measurements follows Equation (Equation 15) and serves as input to Octomap ROS package [40]. The sensors readings are synchronized by zero padding and approximate time algorithm to provide a fair result (see also Figure 10).

## 3. Architecture of Proposed Solution

In this section, the description of our 2D/3D map implementation is discussed. We run our mapping framework using Bayes theorem to update the 3D voxels and store the map as Oc-tree data. In order to map the environment while driving around, a localization system is required. A 2D SLAM framework is used to make a 2D grid map and provide pose estimation as in [16]. The authors use a Rao-Blackwellized particle filter [42] with improved proposal introduced by Doucet in [43] and adaptive sampling for estimating the joint posterior of the map and the position. Rao-Blackwellized technique splits the SLAM problem into two separate position posterior and map posterior:(25)$$p\left(x_{1:t},m|z_{1:t},u_{1:t-1}\right)=p\left(m|x_{1:t},z_{1:t}\right)\cdot p\left(x_{1:t}|z_{1:t},u_{1:t-1}\right).$$

An improved version of proposal distribution is introduced by using scan-matcher to determine the meaningful area in the position likelihood function [43]. Then the prediction step only takes sample in this meaningful area. This reduces the amount of particles needed for sampling the motion model when we have a peak measurement model (e.g., with a LiDAR). After obtaining the robot’s position, mapping with known pose is applied to estimate the 2D occupancy grid. The diagram in Figure 11 shows how the data flow from both sensor through our mapping framework.

First, the 2D laser scanner data is used in a SLAM framework for estimating the position and the 2D map. The achieved localization is also used for the 3D mapping. This is optimal when the sensor measurement from laser is strongly more precise than odometry and the ultrasound. Both inverse sensor models from laser and ultrasonic sensor for updating the grid are synchronized (Figure 10). After that, Bayes filter is applied for updating each voxel in 3D space. The maps considers each grid (or voxel) as binary random variable with discrete states free and occupied. Therefore, the occupancy state will only be updated after several measurements when the probability reach the clamping threshold.

Oc-tree data structure is used to store the map [40]. Making the fused 2D map involves projecting the height data from the 3D map onto x-y-plane. As a result, we achieve a high precision 2D grid map, a more informative fused 2D grid map and a fused 3D map. Figure 12 illustrates the update scenarios based on Equation (Equation 15) as follows:At point (1) and (2): the ISM of US returns p=0.7 ($logodd_{occupied}=0.35$). No information is acquired from the laser. The voxel probability will be updated after each measurement.At point (3): ISMs from both sensors return p=0.7 ($logodd_{occupied}=0.35$). The voxel occupancy probability increases each time there is the measurement from one sensor. If both sensors report occupancy in one measurement, the voxel occupancy will be high even after one scan (p=0.83, $logodd_{occupied}=0.7$).

## 4. Results

The mapping results in 2D plane and 3D space will be discussed in this section. Finally, we propose how the ultrasonic can contribute to 2D navigation and 3D tasks. An environment is established as shown in Figure 13, where there are multiple objects that could be interesting:A human mannequin lying on the ground that can only be partly detected by a 2D laser scanner.An arrangement of tables that contains multiple objects in z-direction that can not be detected by the laser scanner.Several boxes and a trash can that are also out of sight from the laser scanner.Mixed glass/non-glass doors.

### 4.1. 2D Mapping

2D grid maps are used mostly in 2D navigation purposes, where the robot has to find a way to navigate through an environment. With 2D map, we could understand roughly how the environment looks like.

First, a conventional approach is showed in Figure 14a. This results in a malfunction map using only raw ultrasonic measurements for SLAM. Scan matching fails in this case and the robot uses only odometry for localization. Figure 14b shows the raw projected 2D map using the US and laser for localization. We can see the appearance of objects in the grid map which are not discovered by the 2D laser scanner. However, we can also see the unknown (grey area) in the middle. This is the result of sparseness and short-range in measurement. Furthermore, the occupancy grid is updated from the robot base till the end of the measurement beam, where the hit grids contribute to the increment of probability, grids along the measurement beam contribute to the decrements.

Seconly, differences in conventional 2D laser map with one uses addition US data are compared. Figure 15a uses only the laser scanner for SLAM. Therefore, it has a precise representation of the room. Figure 15b uses Bayes theorem to fuse both sensors, then all the voxels are projected onto x-y-plane. Figure 15b on the other hand shows a noisier 2D grid representation by combining the two sensors reading. We mark some areas which are different from the first map with a red circle. On the left side of both maps, we can not see the whole mannequin (annotated as m) lying on the ground but on the second map it is visible. This holds true as well for the trash can (annotated as c) on the top left of the maps. However ghost objects (annotated as g) are discovered through the second map. Last but not least, the table (annotated as t) arrangement is much denser. This indicates a no-go area for a large footprint robot like ours, which only can be seen in the second map. This will contribute a lot for the later planning and navigation tasks.

### 4.2. 3D Mapping

3D maps are important for robot 3D tasks and planning. The map can provide the system a deeper understanding of the surrounding by providing volumetric information. With only the 3D ultrasonic sensor, the environment can be mapped as shown in Figure 16. This 3D representation shows us clearly the objects of interest in the environment. With the additional data from a light weight 3D ultrasonic sensor, the information about the surrounding is much richer. This gives a better input for planning or perception.

However, the 3D space can be improved by adding information of the 2D laser scanner. Applying our method resulting in a wider and more precise space. Figure 17a,b shows a map using the fusion method discussed above. We can see the map’s border is extended and the room shape is much more concrete in our fused map thanks to the laser scanner. The objects of interest are shown in red circles.

## 5. Discussion

The results show that with the help of an affordable 3D ultrasonic sensor, we achieve a more detailed map by making use of Bayes theorem and oc-tree mapping. Without the need of a costly 3D LiDAR, we can still get an abstract representation of the 3D environment. Thanks to the sensor fusion method and the improved localization stack from a 2D SLAM system with the use of laser scanner, we can successively combine both sensors to create both 2D/3D maps. A 2D map is necessary for 2D navigation and planning. Meanwhile, a 3D map is needed for 3D navigation, manipulation tasks and surroundings perception. Oc-tree map representation is used for memory compression and multi-map level resolution. One drawback is that due to the short range measurements of the ultrasonic sensor, we need to drive the robot pretty close to the obstacles in order to achieve a dense 3D map. However, to some extend the 3D fused map can to further works and application like using the manipulator arm of the robot.

In addition, the laser sensor may not work properly in some special circumstances. For example in smoky surroundings, the laser beam is scattered by a high amount of smoke particles as mentioned in [44]. Also in this work, the usage of both laser and ultrasonic sensors have been proved to play an important role for improving decisions in such an environment. With the help of a light weight 3D ultrasonic sensor, this could return a much more promising result.

Needless to say, there are a lot of aspects for future work. We can have two probability pooling methods working together to achieve the best result. Only the z-plane can be updated using Bayes filter. In the meanwhile, the x-y-plane can be updated using weighted pooling methods, namely LIOP or LOP. Changing the sensor pre-processing parameters or the ISM of the 3D ultrasound is also a possible improvement. Noises and ghost measurements from the ultrasound by a better processing mechanism. In addition, the ISM from ultrasound could be modeled to be more precise, so that the associated uncertainty in ultrasound can be addressed. Furthermore, a future evaluation of a full 3D-mapped room with higher precision and resolution is always possible.

## 6. Conclusions

The aim of the present research was to examine a way to fuse both discussed sensors. After an extensive literature review of many current state-of-the-art researches on the topic, including data fusion methods from low level to high level data to map representation, a novel architecture for sensor fusing is described. The fusion approach is compared using simulation. The results of the final map representations, therefore, confirm that it is advantageous to provide any mobile vehicle an additional 3D US sensor to improve its perception.

## Figures and Tables

**Figure 1 sensors-21-03184-f001:**
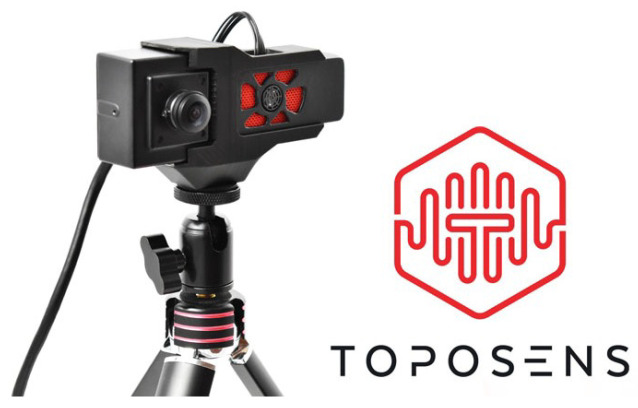
Toposens TS Alpha 3D ultrasonic sensor [12].

**Figure 2 sensors-21-03184-f002:**
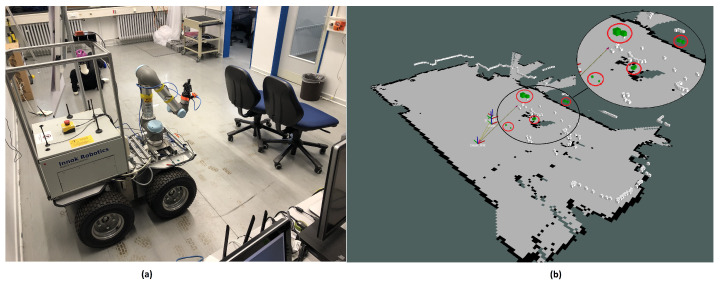
(**a**): Real environment and (**b**): Ground truth LiDAR (white boxes) and 3D US data (green boxes). For convenience, the occupancy grid map using 2DSLAM [16] is included. The LiDAR cannot detect chair’s legs and far corner marked as red circles in (**b**).

**Figure 3 sensors-21-03184-f003:**
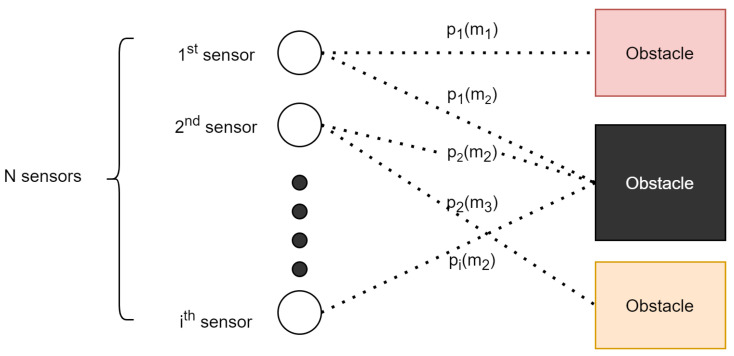
Multiple sensors scenario.

**Figure 4 sensors-21-03184-f004:**
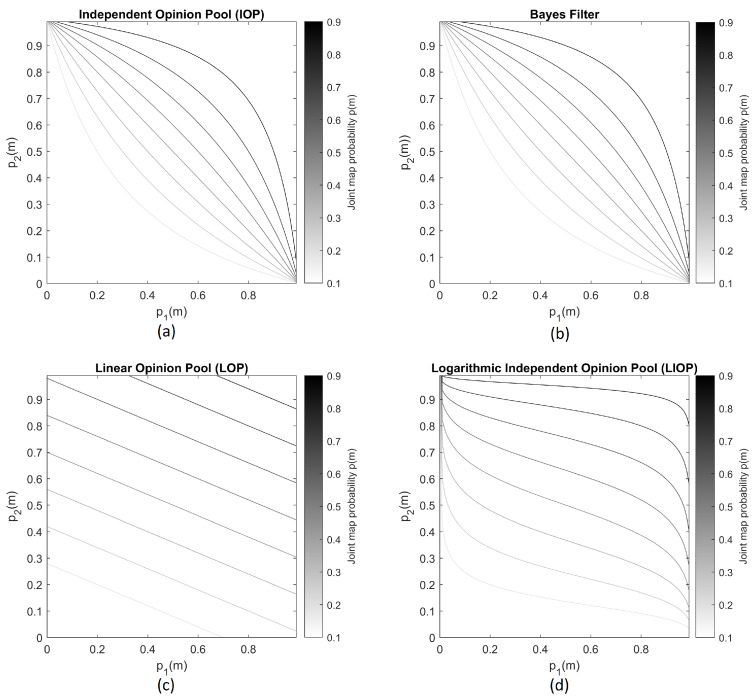
Contour map of p(m) w.r.t. different $p_{1}\left(m\right)$ and $p_{2}\left(m\right)$ (**a**): Independent Opinion Pool (**b**): Bayes filter (**c**): Linear Opinion Pool ($w_{1}=0.4$, $w_{2}=1$) (**d**): Geometric Opinion Pool ($w_{1}=0.25$, $w_{2}=0.75$).

**Figure 5 sensors-21-03184-f005:**
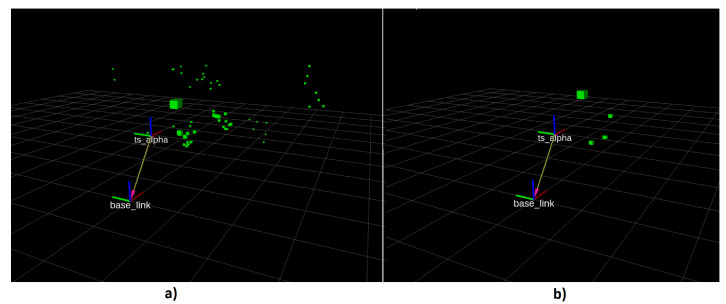
(**a**) 3D ultrasound raw data (**b**) 3D ultrasound after being processed.

**Figure 6 sensors-21-03184-f006:**
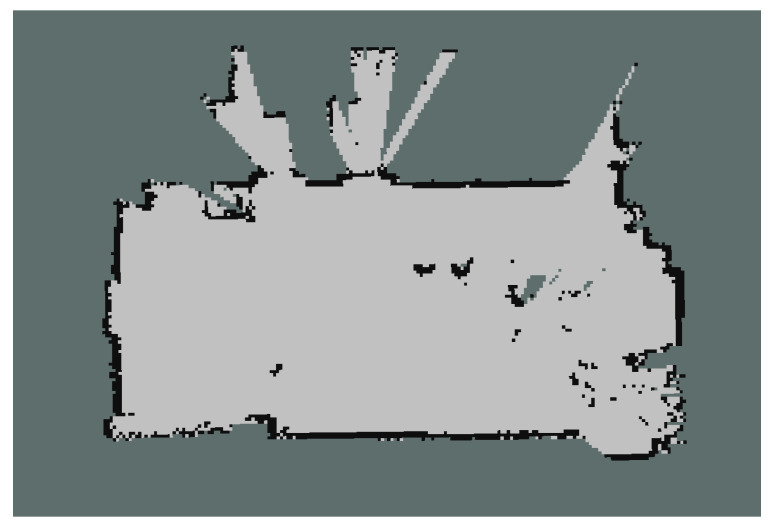
Two-dimensional Occupancy Grid using Gmapping.

**Figure 7 sensors-21-03184-f007:**
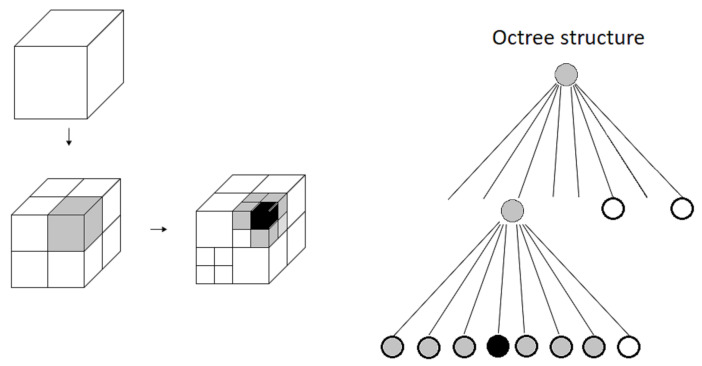
Example of an octree storing free (shaded white) and occupied (black) cells. The volumetric model (**left**) and the corresponding tree representation (**right**). Reproduced from [40].

**Figure 8 sensors-21-03184-f008:**
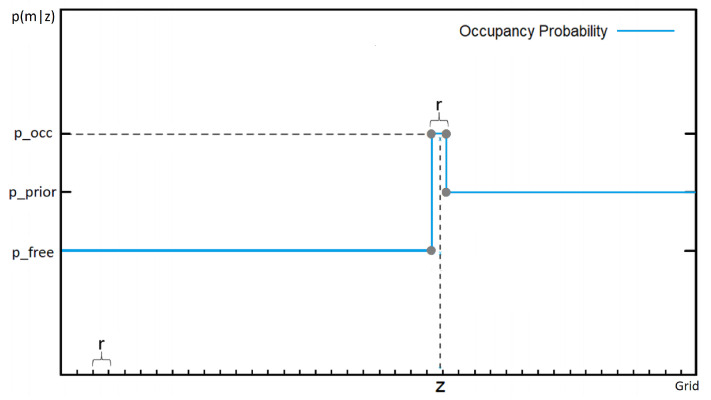
General two-dimensional laser inverse sensor model (*r* = grid cell size, *z* = laser measurement). Reproduced from [34].

**Figure 9 sensors-21-03184-f009:**
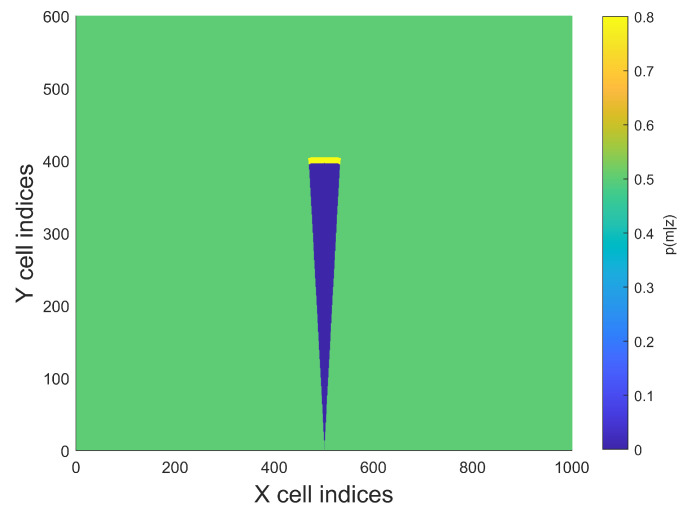
General three-dimensional inverse sensor model.

**Figure 10 sensors-21-03184-f010:**
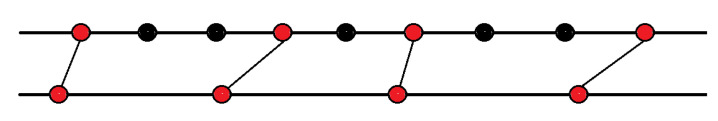
Message synchronization by approximate time algorithm and zero padding. The red dots indicate the taken measurements, black dots indicate the opt out measurements and do zero padding to the merge message in between.

**Figure 11 sensors-21-03184-f011:**
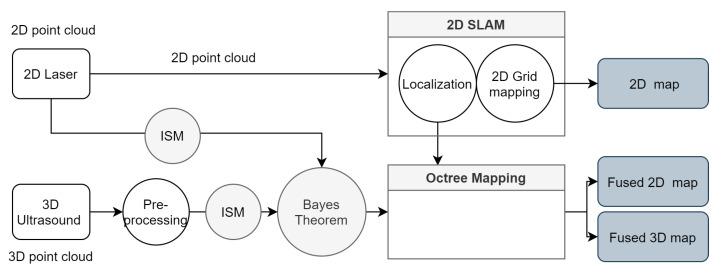
Data flow diagram.

**Figure 12 sensors-21-03184-f012:**
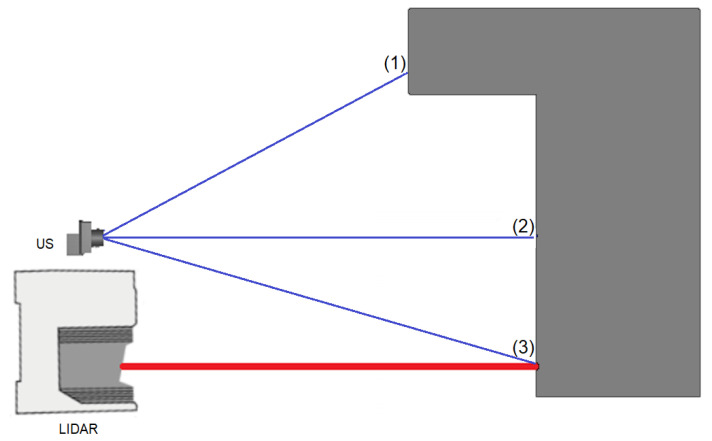
Sensor measurement in different scenarios.

**Figure 13 sensors-21-03184-f013:**
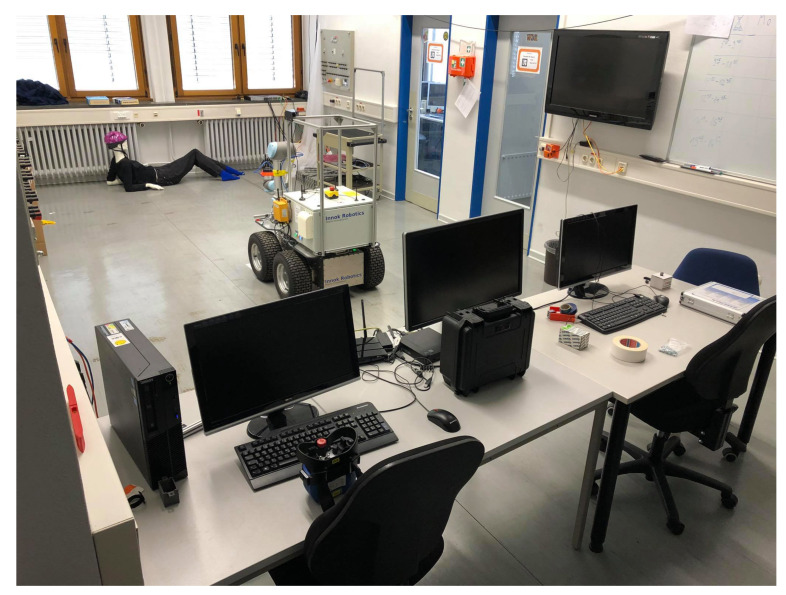
Environment.

**Figure 14 sensors-21-03184-f014:**
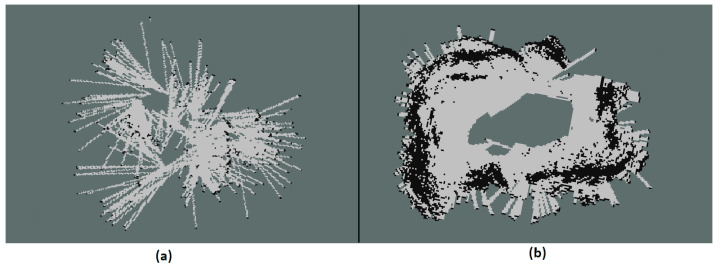
Two-dimensional Occupancy Grid used: (**a**) Raw projected ultrasound measurements for SLAM (**b**) Raw projected ultrasound measurements for mapping and laser scanner for SLAM localization stack.

**Figure 15 sensors-21-03184-f015:**
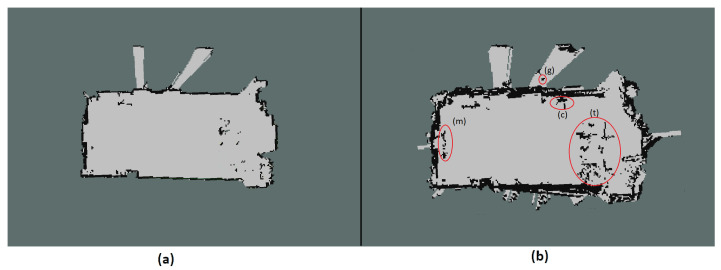
Two-dimensional Occupancy Grid used: (**a**) 2D laser scanner for mapping. (**b**) Fused sensor data for mapping.

**Figure 16 sensors-21-03184-f016:**
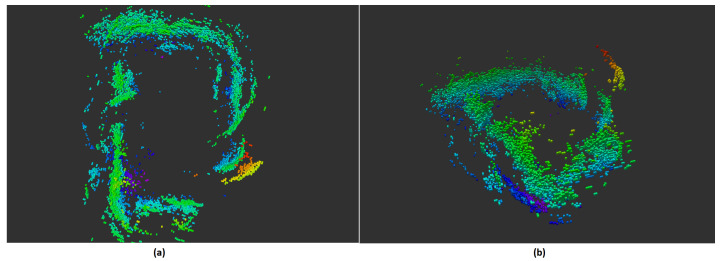
Three-dimensional map used only ultrasound sensor with color represents the voxel’s height (**a**): top view (**b**): side view.

**Figure 17 sensors-21-03184-f017:**
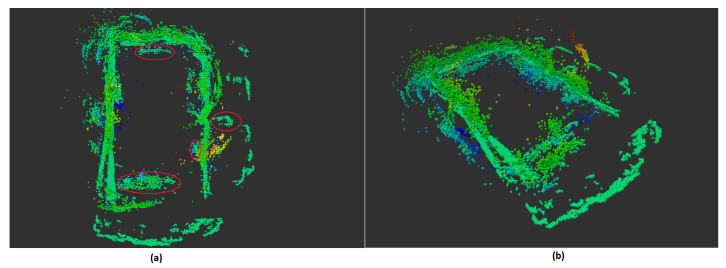
Bayesian approach on two sensor using 3D Oc-tree representation with color represents the voxel’s height (**a**): top view (**b**): side view.

**Table 1 sensors-21-03184-t001:** Comparison between LMS100 and TS Alpha sensors.

	Resolution/Range	Dim.	Refresh Rate	Noise Variance
LMS100 LiDAR	• 0.25°/0.5° in 270° FOV • 20 m range	2D	25 Hz/50 Hz	Low
TSAlpha Ultrasound	• 200 pts in ± 90° azithmuth and altitude • 4 m range	3D	30 Hz	High
	**Noise sources**	**Weight**	**Relative cost**	**Size (L × W × H)**
LMS100 LiDAR	• Reflection material (light absorbing etc.) • Transparent objects	1.1 kg	×12 times	105 mm × 102 mm × 152 mm
TSAlpha Ultrasound	• Reflection material (Soft surface etc.) • Environmental noise • Corner	0.02 kg	×1 time	60 mm × 30 mm × 12 mm

**Table 2 sensors-21-03184-t002:** The six I/O modes for fusion process characterization.

I/O Mode	Characteristics
Data in—data out [17,31]	Data level fusion
Feature in—feature out [17,21,22,23]	Feature level fusion
Decision in—decision out [17,23,32,33]	Decision level fusion
Data in—feature out	The inputs are the raw sensor data and the output are the features of the surroundings.
Data in—decision out	Input of this mode is raw data from multiple sensors, output will be the final decision such as occupancy, objects.
Feature in—decision out	After features extraction from sensors such as camera, the fusion unit will provide the decision as output. Segmentation and object recognition is the example of this fusion mode, where features from multiple sensors will results in a labeled object.

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
