# Peer review of "Environment Mapping Using Sensor Fusion of 2D Laser Scanner and 3D Ultrasonic Sensor for a Real Mobile Robot"

_sensors, 2021, doi:10.3390/s21093184_

Round 1
Reviewer 1 Report
This manuscript presents a method for fusing 2D laser measurements with those of a 3D ultrasonic sensor. The method is applied for situational awareness and mapping for a mobile robot. The results show that this method gives much better results compared to using only 2D laser, and the potential to generate 2D/3D maps without the need for expensive 3D lasers.
Literature overview, equipment descriptions and sensor fusion method used, as well as the results are presented with great clarity and a good flow.
Below are some remarks given by the Reviewer to the Authors with the goal to further improve their manuscript:
1. "1.1 Outline" - Reference the Sections by numbers and hyperlink.
2. Change "we present/..." form into the indirect "... is presented/..." form throughout the manuscript.
3. Line 76: Put toposense.com link as a misc source in the reference list.
4. Add specs and details about the mobile robot and the sensors. It would even be interesting to compare these sensors w.r.t. some more expensive 3D laser scanners and their mapping results.
5. Figures 8, 9, 12. - The resolution should be increased to at least 300dpi.
6. Figure 15b & lines 390-401: Annotate characteristic objects to improve flow and readability.
7. Sections 4.1. & 4.2. Offer some sort of quality metric estimation w.r.t. the ground truth 2D/3D model and/or 3D laser-recorded room (if available) in order to evaluate the improvement of the proposed method in a simple and objective way, or at least discuss the possibility in Section 5.
8. Remove DOIs from the References.
Reviewer 2 Report
The manuscript titled - " Environment Mapping Using Sensor Fusion of 2D Laser Scanner and 3D Ultrasonic Sensor for A Real Mobile Robot " presents a methodology to fuse the 2D laser scanner and 3D ultrasound for 2D/3D representation of the surrounding based on Bayesian Filter. The detailed explanation of the sensors, and related building blocks are presented. The manuscript can be accepted, provided that the following is taken into account
Authors move back and forth between state of the art and the methods. This impedes the flow of method and is not clear what the authors claim to achieve with their methods ?
Further, authors should work on identifying the building blocks of the method and re-arrange the content.
The author explains the mathematics and theory behind the implementation, though the equations and its implications are poorly explained. This should be explained in detail.
I would recommend the author to connect the different constituents parts such as - sensor (section 2.1 ), sensor fusion (section 2.2, 2.3) map representation (section 2.6) in the form of block diagram. This will enhance the readability and add over all work flow.
Authors should include the algorithm explaining how different aspect - such as sensor fusion, map gridding and updating .. are performed
Minor issues:
Line 39 - 'After that, we also study main ..... Here, I guess authors wants to present a comparison. So 'method' after comparison must be removed.
Line 74 - The author mentions about high resolution, precision and range of SICK laser. What are those numbers ? Adding them would add value to the manuscript. Also the word these numbers are relative so 'high' should be removed.
Line 86 - esp. should be changed to especially.
Through out the manuscript, the figures should be either placed at the top or end.
What is the difference between p(m) (eq. 1,2) and p_i(m_i) in Fig. 3 ? For the ease of understanding how these eq 1, 2 can be expressed in terms of p_i(m_i) ? should be mentioned before Fig. 3 (though eq. 4 explains the relationships)
Line 160 - i.e.
line 190 - noisy and sparse
line 191 - We draw the output of computed p(m) does not make sense. May be rephrase the sentence
Fig. 4 should be replaced with high quality graphs. Further p1(m) should be changed to p_1(m) and same for other parameters. Same with Fig. 9
line 267 ... and thus not applicable to represent 3D environment. This statement is not true. Though the memory and computational cost are higher, they can be used for 3D mapping.
line 274 - octree and similar at several instances
Authors should explain how ISM is obtained ? It is not clear on what basis the author chose p_free and other parameters ?. Further the implications of the parameters should also be explained.
line 302 - Fig 8 doesn't have (a) ?
line 348 - where is the ISM for US ? and or how it is obtained ? It is missing
Reviewer 3 Report
Authors proposed the environment mapping using 2D laser and 3D ultrasonic sensor for mobile robot. That work is first fusion work to improve the reliability of visualized scanning models. Authors proved the concept using mathematical analysis and measured results and showed the effectiveness of the proposed idea. There are no grammar mistakes so the manuscript looks smoothly understood for readers. This is well written manuscript to show the effective ultrasonic and laser scanner with a higher and more accurate results. However, authors need to add some missing references and previous paper reviews. Therefore, the manuscript could be minor revision if authors follows the comments.
1. In the reference section, authors should use abbreviated journal names.
2. In Figure 9, there are too small size letters with unclear fonts.
3. In Figure 8, letters are not clear to be seen.
4. In Figure 5, please use larger fonts such as base_link.
5. In Figure 4, it is hard to see the values of several m.
6. Please describe the advantages and disadvantages of the previous work of ultrasonic detection methods if possible in the introduction section (1.2. Related work). Authors mentioned that there are no 3D ultrasonic sensor platform. However, there are some patent work. If authors cannot find that, it is fine to leave that.
7. Please add the reference (Due to physical characteristics of sound, the resulting data is effected strongly by surrounding noise or the material of the targets.) with the reference (Kim, K., & Choi, H. (2021). High-efficiency high-voltage class F amplifier for high-frequency wireless ultrasound systems. Plos one, 16(3), e0249034.)
8. Please add the reference (Another disadvantage of the sensor is that it can not detect walls as lines in space due to the dominant specular reflection) with the reference (Wei, Teng, Anfu Zhou, and Xinyu Zhang. "Facilitating robust 60 ghz network deployment by sensing ambient reflectors." 14th {USENIX} Symposium on Networked Systems Design and Implementation ({NSDI} 17). 2017. ) or another reference.
9. what is red and black circles in Figure 11 ? Please describe in detail.
10. what is p in Equation (2) ? p is probability ?
